# Environmental Surveillance of Legionellosis within an Italian University Hospital—Results of 15 Years of Analysis

**DOI:** 10.3390/ijerph16071103

**Published:** 2019-03-28

**Authors:** Pasqualina Laganà, Alessio Facciolà, Roberta Palermo, Santi Delia

**Affiliations:** 1Regional Reference Laboratory of Clinical and Environmental Surveillance of Legionellosis, Branch of Messina, Department of Biomedical and Dental Sciences and Morphofunctional Imaging, University of Messina, Torre Biologica 3p, AOU ‘G. Martino, Via C. Valeria, s.n.c., 98125 Messina, Italy; adelia@unime.it; 2Department of Clinical and Experimental Medicine, University of Messina, 98125 Messina, Italy; afacciola@unime.it; 3Department of Biomedical and Dental Sciences and Morphofunctional Imaging, University of Messina, 98125 Messina, Italy; r.palermo91@gmail.com

**Keywords:** *Legionella*, legionellosis, environmental surveillance, water system

## Abstract

Legionnaires’ disease is normally acquired by inhalation of legionellae from a contaminated environmental source. Water systems of large and old buildings, such as hospitals, can be contaminated with legionellae and therefore represent a potential risk for the hospital population. In this study, we demonstrated the constant presence of *Legionella* in water samples from the water system of a large university hospital in Messina (Sicily, Italy) consisting of 11 separate pavilions during a period of 15 years (2004–2018). In total, 1346 hot water samples were collected between January 2004 and December 2018. During this period, to recover *Legionella* spp. from water samples, the standard procedures reported by the 2000 Italian Guidelines were adopted; from May 2015 to 2018 Italian Guidelines revised in 2015 (ISS, 2015) were used. Most water samples (72%) were positive to *L. pneumophila* serogroups 2–14, whereas *L. pneumophila* serogroup 1 accounted for 18% and non-*Legionella*
*pneumophila* spp. Accounted for 15%. Most of the positive samples were found in the buildings where the following critical wards are situated: (Intensive Care Unit) ICU, Neurosurgery, Surgeries, Pneumology, and Neonatal Intensive Unit Care. This study highlights the importance of the continuous monitoring of hospital water samples to prevent the potential risk of nosocomial legionellosis.

## 1. Introduction

The *Legionella* genus currently includes 61 saprophytic species and about 70 serogroups (sgrs), of which at least 20 have been recognized as responsible for opportunistic infections in human beings. The most frequently isolated species from cases of infection is *L. pneumophila*, which is subdivided into 16 different sgrs, of which the serogroup (sgr) 1 is the most pathogenic and causes 70% to 90% of all cases of legionellosis [1,2,3,4]. In contrast, *L. pneumophila* sgrs 2–14 account for only 15% to 20% of community-acquired cases, although they account for over 50% of the isolates obtained from man-made aquatic systems [5]. In Italy, since 1990 legionellosis has been included among infectious diseases with compulsory notification (Class II: major diseases due to high frequency and/or subject to control actions) [6]. When a new case of legionellosis is diagnosed, physicians notify the local health authority that subsequently informs the regional and consequently national authorities. 

The number of cases of legionellosis has increased steadily over the years, both in Europe and in Italy, in the latter due mainly to the numerous receptive structures (hotels, spas, etc.) present in the territory [7]. In 2015, in Italy, most cases were community acquired (78.8%), followed by travel-associated (12.7%) and healthcare-associated (5.3%) cases. Particularly, in Sicily, in the last fourteen years (2004–2017) 143 total cases were reported. The cases verified in 2018 have not been published (Annual Reports of Legionellosis, Istituto Superiore di Sanità) [8]. 

In hospitals and other healthcare facilities, waterborne diseases may originate from the bacterial colonization of water pipes, taps, cooling towers, showers, and water supplies [9,10]. Moreover, previous studies investigated the role of unusual sources of *Legionella* in hospitals, such as bubblers for oxygen-therapy, pediatric incubators, dental chairs, etc. [11], in closed community environments, such as schools and prisons (data not published), beach showers, and cruise ships [12,13]. 

Periodic monitoring of the presence of *Legionella* in the water network allows preventive action to be taken to avoid possible exposure of patients and health professionals [14,15].

*Legionella* is able to survive for long periods in water and even to replicate in the presence of chlorine, if it manages to create suitable conditions (areas of stagnation and sludge formation, parasitism of amoebas and protozoic cysts, etc.) [8,16]. Water systems represent suitable environments for the growth and multiplication of *Legionella* spp. and other Gram-negative bacteria, which survive at different pHs and temperatures [17,18,19,20,21]. 

The risk of illness increases dramatically if the germ is found in certain wards such as intensive care units, hematology-oncology units, cardiology units, hemodialysis units, and pulmonology units due to the critical nature of these wards for their hospitalized patients [22,23,24,25,26,27,28].

The interest of our laboratory in *Legionella* was born in 1988. At first, it was involved only in research activities aimed at recovery of *Legionella* in the small hospitals, care homes, and other types of structures in Messina. Since 2004, it began to carry out continuous surveillance at the University Hospital “G. Martino” and in 2012, was appointed the Regional Reference Laboratory of Clinical and Environmental Surveillance of Legionellosis, branch of Messina (Italy), situated within the Polyclinic area. 

The purpose of this paper is to present a retrospective study of pluri-annual environmental surveillance conducted by our laboratory to evaluate the potential risk for hospitalized patients and healthcare workers to get sick of legionellosis.

## 2. Materials and Methods

In this paper, we present the results of surveillance conducted for 15 years, from 2004 to 2018. As mentioned earlier, environmental monitoring of *Legionella* in the water pipelines within the University Hospital in Messina began in 1988. After a few years of desultory analysis, we transitioned into a real continuous environmental surveillance with constant monitoring of all the pavilions that make up the structure. Monitoring is also active now. 

### 2.1. Hospital Structure

The Messina University Hospital is a large structure that extends over an area of about 310,000 m^2^ and it is distributed on 11 pavilions, indicated by letters of the alphabet (Figure 1), which have in total four to six floors raised and which house various university- and hospital-related services and departments (such as clinics, hospital rooms, operating rooms, laboratories, changing rooms for doctors and nurses, kitchens and services, classrooms, libraries, refectory, etc.). The construction of the first pavilion dates back to 1967, but it is an ever-evolving structure with expansions and modifications dictated by the requirements of adapting to the most modern welfare standards.

### 2.2. Sampling

From January 2004 to December 2018, 1346 samples were collected from the water distribution system of the “G. Martino” University Hospital (Messina, Italy) and were examined for *Legionella*. Monthly samplings were performed in 11 different buildings hosting the various wards. The groundwater supplied to the hospital is provided by the municipality and is disinfected with chlorine dioxide; the water reaches the hospital by means of a single pipeline that leads to a centralized tank where the water is stored. The water does not undergo further chlorination after it is gathered from the Messina town pipeline. From the centralized tank, the water is distributed to each building by electric-motor pumps that send it through a pipeline that runs across the basements of all the buildings. Under each building there is a boiler that produces heated water (average temperature approximately 43–48 °C) that climbs up again to supply the wards located on each floor. Samples of heated water were collected at the start of daily activities from taps, without running the water, in accordance to the 2000 [29] and 2015 [3] Italian Guidelines, using 1-L sterile glass bottles with 1% sodium thiosulfate to neutralize the presence of chlorine. In all the sampled points the water temperature was measured with a precision thermometer (Temp-16 RTD Thermometer, ThermoFisher Scientific, Waltham, MA, USA) immersed in the flow of running water. 

Water samplings were carried out constantly, continued over time, about two times a month. In order to obtain a sampling representative of the hygienic-sanitary conditions, care was taken to sample all the floors, and the wards located both on the left and on the right sides inside each building. Particularly, we collected the water samples from taps of bathrooms located inside and outside the wardrooms (i.e., aisles, waiting rooms), clinics, laboratories, operating, sterilization, storage, and medical/nursing rooms. The most sampled sites were taps of bathrooms situated outside the wardrooms (29%) and clinics (28%), followed by taps of wardrooms (17%) and laboratories (14%). The collected samples were carried to the laboratory within 30 min and processed shortly afterwards—no further precautions were taken to prevent sample contamination.

### 2.3. Isolation and Serological Identification of Legionella Spp.

From January 2004 to April 2015, to recover *Legionella* spp. from water samples the standard procedures reported by the 2000 Italian Guidelines were adopted; from May 2015 to 2018 the Italian Guidelines revised in 2015 were used. In the last years, in accordance to the guidelines, 1-L water samples were concentrated to 10 mL through 0.2-μm porosity membrane filters and incubated at 50 °C for 30 min in a thermostatic bath. Concentrated and unconcentrated samples were spread on duplicate plates of Buffered Charcoal-Yeast Extract Agar Base Medium (BCYE, Oxoid Ltd., Basingstoke, United Kingdom) and incubated for 10 days at 36–37 °C in a moist chamber with 2.5% CO_2_. The suspected colonies were isolated and confirmed as *Legionella* spp. after screening their inability to grow on a culture medium without cysteine. *Legionella* counts were reported in colony forming units/liter (CFUs/L) according to the number of colonies per plate and to the dilutions performed on the original sample. A latex micro-agglutination test kit with polyvalent antisera (BCYE, Oxoid Ltd., Basingstoke, United Kingdom) was used to identify the isolates assumed to belong to *Legionella* genus. For serological identification also “*Legionella pneumophila* monovalent antisera set 1 and 2” and *Legionella* antisera for several non-*L. pneumophila* spp as *L. bozemanii*, *dumoffii*, *gormanii*, *micdadei*, etc. (Denka Seiken co. Ltd., Tokyo, Japan) were used. Concerning the bacterial load, we divided the isolates into four groups according to the latest national guidelines (modified from the point 3.4: “Risk Assessment and Management in the healthcare facilities) [3]:1 = 101–1000 CFU/L;2 = 1001–10,000 CFU/L;3 = 10,001–100,000 CFU/L;4 > 100,000 CFU/L.

Samples with a bacterial load lower than 100 CFU/L were not taken into consideration because no intervention was required; moreover, another range was added (>100,000) because some of our water samples significantly exceeded the limit of 10,000 CFU/L of *Legionella*.

## 3. Results

Table 1 shows the percentages of samples collected during the entire period in each building of the studied University Hospital and the percentages of positive and negative samples. The table shows that the highest numbers of water samples were collected in the buildings where the “critical” wards are situated. In detail, the building where we collected the highest number of samples (21%) was building E (ICU and Neurosurgery), followed by the buildings H (Pulmonology, Infectious Diseases, Hematology, Oncology), F (General Surgery), and NI (Pediatrics), with percentages of 13%, 12%, and 12%, respectively. Of 1346 collected water samples, 812 (60%) were positive. Table 1 shows the different buildings, along with the principal medical activities that take place in each of the buildings, and the different serotypes isolated in each building. Of all samples, 42 (3%) were positive to more than one serotype at the same time. Particularly, 9 (21%) were positive to *L. pneumophila* 1 + *L. pneumophila* 2–14; 14 (34%) were positive to *L. pneumophila* 1 + non-*L. pneumophila* spp.; 16 (38%) were positive to *L. pneumophila* 2–14 + non-*L. pneumophila* spp.; and only 3 (7%) were positive to all the tested strains.

Regarding positive water samples, most of them were contaminated by *L. pneumophila* sgrs 2–14 (72%), whereas *L. pneumophila* sgr 1 accounted for 18%. A certain percentage of samples (15%) was positive to non-*L. pneumophila* spp. The total percentage is not equal to 100% because we found different *Legionella* serotypes in the same sample. The buildings where we found a major positivity rate were the E (77%), H (76%), and A (71%) buildings, followed by distance from the others. 

Most of the *L. pneumophila* 1 isolated was in the first two groups (101–1000 CFU/L and 1001–10,000 CFU/L); the rest was in the third. *L. pneumophila* 2–14 isolates had a load belonging for the most part to the first three groups; only a small number of isolates belonged to the fourth group. Finally, non-*L. pneumophila* spp. isolates were primarily found in the first group, with a similar number of cases observed in the second and the third group (Figure 2).

Figure 3 shows the positivity trend during the considered period. The figure shows a rather constant presence of positive samples during the considered years, but a decreasing trend is evident in the most recent years. Figure 4 shows the trend of the different isolated *Legionella* serogroups. 

## 4. Discussion

Legionellae are microorganisms that have their natural habitat in the water and can easily reach pipelines, where they can reproduce via the presence of biofilm in the inner walls of the pipes. In hot water distribution networks, often the temperature coincides with the optimum temperature for the replication of these bacteria. Further favorable factors are the lengths and the tortuous formats of the hot water distribution networks normally present in big buildings like hospitals. 

Building via continuous renovations could promote the presence and the growth of *Legionella*. Moreover, the restructuring of buildings often does not involve replacing the water pipelines, which typically remain those of origin. For all these reasons, it is not surprising that levels of significant contamination have been found in most of the healthcare-related structures where similar surveys were conducted [30,31,32]. The presence of fouling in the pipes and dead points with stagnation of the water in the networks well explain the limited effectiveness of water chlorination and heat treatments. Indeed, the eradication of legionellae contamination in hospitals is very difficult due to the common failure of the decontamination interventions of the water networks [33,34]. One possible solution to prevent the spread of *Legionella* from water is the application of filters to taps, thus allowing for germ-free water, although this involves a certain economic cost.

In this study, our surveillance demonstrates the constant presence of *Legionella* strains in the water supplied in a big hospital structure over a long period of time (15 years). The surveillance was carried out especially in the buildings where critical wards (such as those of the ICU, neurosurgery, and oncology wards) are located. The more important issue to highlight is that these buildings were those where we found the highest positivity rate. In these wards, the inpatients are often immunocompromised, and thus they are consequently at high risk of contracting many infections. This result shows that these patients have a potentially high risk of contracting legionellosis with the subsequent worsening of their already compromised health conditions. This issue is especially true concerning inpatients of ICU and neurosurgery wards, hospitalized in the building E, where we found a large presence of *L. pneumophila* 1 (52%), which is known to be the most pathogen strain of the genus. However, as shown in Figure 2, for this strain we found a bacterial load lower to 10,000 CFU/ml.

It should be noted that whenever the bacterial load exceeded the limits defined by the national guidelines, a hyper-chlorination intervention was carried out both on the entire water system and on every single positive faucet. Furthermore, at each single sampling point, the final part of the tap was replaced and, in the most critical areas, an anti-*Legionella* filter was applied. Indeed, this is likely the reason why the trend calculated over the considered period has a non-linear tendency, with peaks and valleys, although it may also be partially resulting from the seasonality of the sampling.

To prevent nosocomial cases of legionellosis, national and international guidelines recommend measures to control *Legionella* water contamination, with particular reference to healthcare structures [30,35]. In particular, for *Legionella* control it is important to focus attention on three basic parameters: (1) the amount of the germ in the water, through analytical monitoring; (2) the presence of virulence factors in isolated strains; (3) the receptivity of the guests [36]. As the World Health Organization (WHO) suggests, the best approach to assessing the health risks associated with exposure to *Legionella* is the development of a water safety plan (WSP), which is an operational tool that is useful for the systematic assessment or risk analysis of pipelines. Particularly, the plan is a dynamic tool that gives priority to the risks and dangers of pipeline colonization and considers the most appropriate control measures as well as the possible obstacles to their realization.

Despite the environmental surveillance having been constant and thorough throughout the 15 years of study, the parallel clinical surveillance was, on the contrary, poor and limited to a few urine samples. The wards that proved to be more careful in regard to this issue were the ICU ward, pneumology ward, infectious diseases ward, and internal medicine ward. However, despite this attention, the commitment to this survey is still far from being satisfactory. We can deduce that there is still a long way to go. This issue is a sign that there are likely still a remarkable number of undocumented cases of *Legionella* infections present even among healthcare workers. Probably, physicians are not used to thinking of *Legionella* as a possible cause of nosocomial pneumonias. It is necessary for continuous education to be provided to the healthcare workers in order to improve the situation, to understand the real incidence of legionellosis, and, especially, to obtain a correct diagnosis and treatment of this infection.

## 5. Conclusions

This paper emphasizes the need to develop a risk analysis aimed at controlling and preventing *Legionella* in all community structures. In particular, through the description of the surveillance actions carried out, the Regional Reference Laboratory for Legionellosis of Messina suggests that specific critical control points (CCPs) should be identified according to the logic of Hazard Analysis and Critical Control Point (HACCP). Moreover, due to the evidence of the presence of *Legionella* even in the aerosol, in very recent years we have associated the control of this latter to the routine water control (data not yet published). Indeed, many recent studies suggest and recommend concurrent research on water/air to increase the probability of finding microorganisms [37,38,39]. These aspects are important not only in order to plan the management and monitoring actions of the pipelines, but also in view of a public health approach, with the specific aim of preventing illnesses in the exposed population.

## 6. Limits of the Study

The study is just a presentation of data demonstrating the presence of *Legionella* linked to environmental surveillance that has lasted for years. Clinical data were not analyzed because they are very poor. In the last period, not included in this study, there was a greater propensity by clinicians to increase the research of *Legionella* on human samples—these data will soon be presented in a future study.

## Figures and Tables

**Figure 1 ijerph-16-01103-f001:**
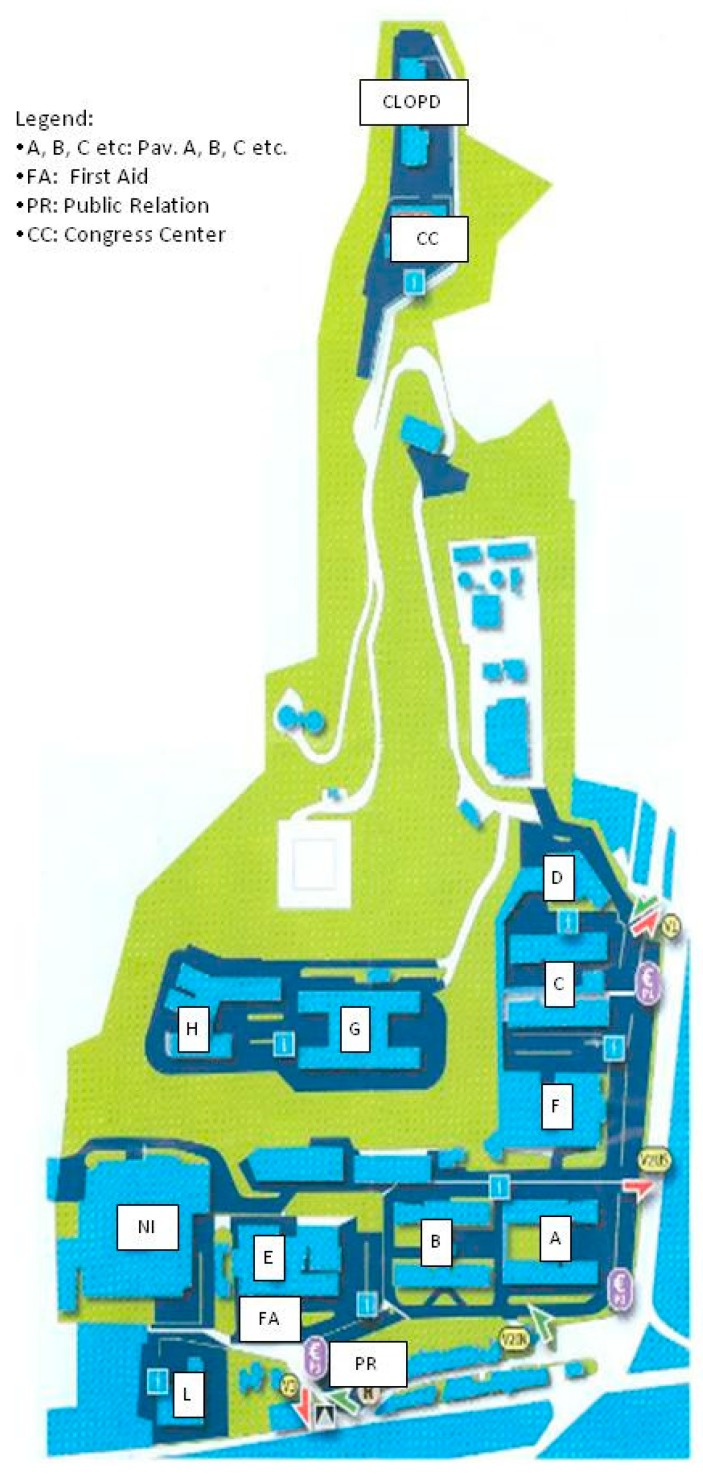
University Hospital of Messina map.

**Figure 2 ijerph-16-01103-f002:**
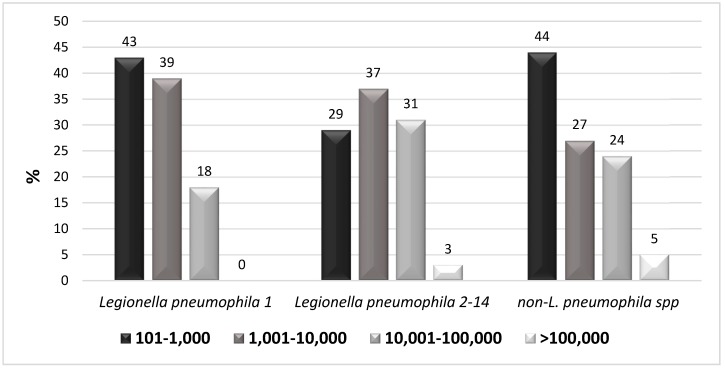
Bacterial load of the different *Legionella* serotypes isolated by water samples.

**Figure 3 ijerph-16-01103-f003:**
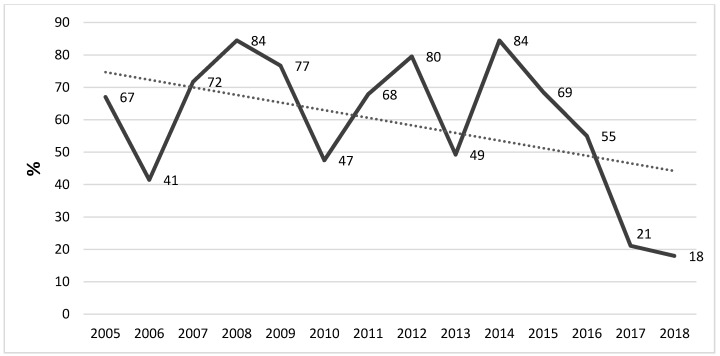
Positivity trend of *Legionella* isolates in the considered period.

**Figure 4 ijerph-16-01103-f004:**
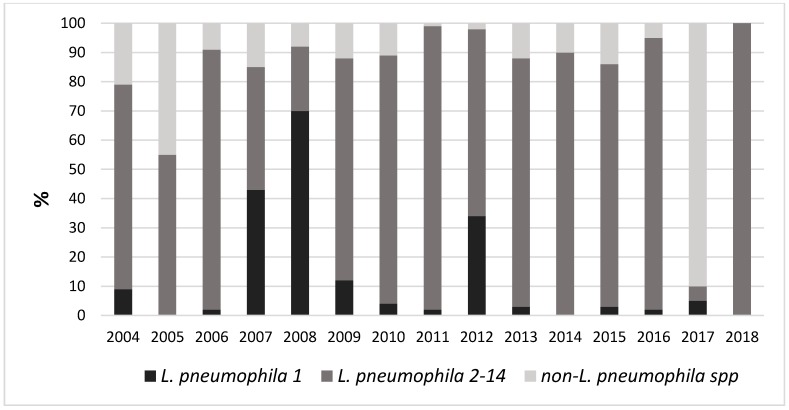
Positivity of different isolated *Legionella* serogroups.

**Table 1 ijerph-16-01103-t001:** The positivity rates and serotypes isolated according to building/principal medical activity.

Building	Wards	Total	Negative	Positive	Positive for *LP* 1	Positive for *LP* 2–14	Positive for Non-*LP* spp.
A	Obstetrics Gynecology	87 (6%)	25 (29%)	62 (71%)	4 (6%)	44 (71%)	17 (27%)
B	Internal Medicine	67 (5%)	29 (43%)	38 (57%)	0	21 (55%)	20 (53%)
C	Cardiology-Nephrology	95 (7%)	68 (72%)	27 (28%)	0	26 (96%)	1 (4%)
D	Pathological Anatomy	76 (6%)	37 (49%)	39 (51%)	2 (5%)	37 (95%)	4 (10%)
E	ICU Neurosurgery-Orthopedics	280 (21%)	64 (23%)	216 (77%)	112 (52%)	101 (47%)	28 (25%)
F	General Surgeries	163 (12%)	69 (42%)	94 (58%)	2 (2%)	88 (94%)	5 (5%)
G	Laboratories	136 (10%)	52 (38%)	84 (62%)	26 (31%)	58 (69%)	2 (2%)
H	Pulmonology, Infectious Dis., Thoracic Surg., Oncology	173 (13%)	41 (24%)	132 (76%)	0	118 (89%)	14 (11%)
NI	Pediatrics	160 (12%)	60 (38%)	100 (62%)	1 (1%)	93 (93%)	10 (10%)
W	Clinics	13 (1%)	11 (85%)	2 (15%)	0	1 (50%)	1 (50%)
CLOPD	Dentistry	96 (7%)	78 (81%)	18 (19%)	0	1 (5%)	17 (95%)
Total (%)		1346	534 (40%)	812 (60%)	147 (18%)	588 (72%)	119 (15%)

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
