# Peer review of "Environmental Surveillance of Legionellosis within an Italian University Hospital—Results of 15 Years of Analysis"

_ijerph, 2019, doi:10.3390/ijerph16071103_

Round 1
Reviewer 1 Report
Please find comments attached.

Author Response
Dear Reviewer,
First of all, we would thank for your comment, we greatly appreciated the suggestions and we have addressed the points according to their request.
Please find in attached file the revised version of the manuscript “ijerph-446008”; the Authors have tried to make the changes in agreement with the referees' suggestions and the reply to each question moved. The Authors hope that in the current version the manuscript is clearer and more suitable to be published in International Journal of Environmental Research and Public Health.

Reviewer 2 Report
Interestingly written article, but I suggest supplementing it with several elements, important especially for practitioners:
(1) it would be important to have more information on the cases of Legionellosis diagnosed in previous years in Sicily: species and serotypes;
(2) what changes have been implemented to prevent the risk of Legionella contamination in the water system of studied hospital?
Author Response

(The authors gave the same response as above.)

Reviewer 3 Report
Strenght
The topic presented in this paper is indeed current and relevant, and the authors apparently dealt overtime with the water contaminated by Legionella with a highly structured system, and highlighted with strong evidence that contamination is a fact, and that the risk it carries in this specific background has to be assessed.
The abstract is thorough and encloses all the information needed to figure out the topic.
The title is clear and representative of the assembly.
English is sufficient and doesn’t need too much reworking.
Weakness
General comments:
There is a generalized mixing up of contents throughout the sections of the article. Authors should take care of removing redundant sentences from inappropriate sections, and also move important information to the proper section.
Introduction:
Consider expanding the section adding what justified this research and enhance the explanation above the purposes.
Lines 46 to 47: Consider changing the sentence to "no nosocomial case was notified".
Lines 48 to 49: Consider citing only 1-2 references for this statement.
Lines 49 to 52: Consider citing only 1-2 references for this statement.
Line 51: Consider changing the expression “communities environment” to "closed or crowded community environments".
Lines 59 to 61: Consider extending this statement by adding a quantitative evaluation of the problem.
Line 62: I recommend the use of the conjunction "in" instead of “to”.
Line 65: Please provide the correct typing for the year which I assumed to be 2012.
Lines 68 to 72: Consider that highlighting the fact that the constant positivity of Legionella over time, may represent a reason for starting this research.
Secondly, consider exposing the purpose just below, being careful not to mix it up with the description of the study design, which should be kept within the "Materials and Methods" section.
Consider, finally, not providing the conclusions here, since it could appear redundant.
Lines 73 to 74: Consider deleting this whole aspect of your research, since neither results were provided nor discussed.
Materials and Methods:
Sampling:
Consider providing more details about the sampling process, whereby applicable.
You could for instance clarify how did you deal with the chlorine residual effects on the representatively of the samples.
Also consider informing the reader whether you were investigating the presence of Legionella inside the pipeline network or at the tap. Please consider describing the measures you took to avoid sample contaminations, especially if the first case is applicable.
Also consider expanding the description about the temperature measurement operations.
Please describe the means of sample tracking you undertake during the procedures; consider specifically reporting the time between sampling and analysis.
Lastly, consider citing a reference which deals with the sampling method followed.
Isolation and serological identification of Legionella spp.:
Lines 88 to 90: Consider specifying if you actually measured the temperature of the water coming from the boilers, or the data provided was reported by others. Is “13” a typo?
Line 104: Consider citing product rights owner (Oxoid Ltd., Basingstoke, United Kingdom)
Line 111: Consider citing the actual product rights owner (Denka Seiken co. Ltd., Tokyo, Japan).
Results:
Needs more attention in data matching between text and figures.
Line 121: The percentage for building E (21%) doesn't match with the corresponding one in Figure 1 (22%).
Line 123: Percentages don't match with those in Figure 1, in which H is 13%, F is 13% and NI is 12%.
Consider also sorting these records coherently with Figure 1 ( the order should be F, H, NI).
Table 1: Consider formatting the headlines in order to make them easier to read.
Lines 132 to 134: Please reword these sentences clarifying that the percentages refer to the positive share of the samples analyzed, ensuring that the reader doesn't relate them to the overall amount of samples collected.
Lines 136 to 138: Consider moving this sentence to section 2.1 (Sampling), since it doesn't carry actual results, but instead it gives thorough information about the sampling process.
Lines 138 to 143: Consider moving this to section 2.2 (Isolation and serological identification of Legionella spp.), and also rewording group 1 to "101-1000 CFU/L" since this is the format dictated by the 2015 guidelines (also do this for Figure 2). Finally, consider citing the reference to the guidelines here.
Lines 155 to 156: Consider not focusing just on the trend, but also on the peaks and valleys, reporting this fact in the text for discussion purposes.
Lines 156 to 157: Consider adding more details about the results reported in Figure 4.
Temporal characterization of Legionella positivity during the given period needs to be expanded. To better highlight possible trend more points could be produced (months instead of year). It seems that the positivity could be related to some events which could be investigated. As I reader I get curious about it.
Discussion:
The results included in figures 3 and 4 need to be discussed as well.
Since this section also states the lack in a quantitative analysis around the impact of the results obtained on health in this specific setting, therefore it resembles forthrightly a bit insubstantial.
At the same time, there is a clear chance to give this paper more impact. This lies in the alternating aspect of Figure 3, which needs to be highlighted and further examined for discussion.
Line 169: Consider the use of the more common term "clinics" instead of “ambulatories”.
Figure 5: Consider deleting Figure 5 since it’s not a key element for the comprehension of the setting.
Line 175: Please add a reference to this statement.
Line 198: I recommend rewording the sentence “it is important to focused the attention” to "it is important to focus the attention" or "it is strongly focused the attention", according to the original meaning you wanted to give this statement.
Lines 206 to 215: Since this part of the research didn't give enough results, I suggest you to delete it. Anyway, please stay coherent with other parts of the paper (including the title) in which you write "15 years" with reference to the period of surveillance.
Conclusions:
Consider reconnecting with the purpose of this research, stating whether it has been achieved or not, and motivating it. Moreover, I strongly suggest you to declare the limits of this study.
Actually this section is coherent with the whole text and it’s going directly to the point, but, also in view of the above, it doesn’t add so much of remarkable to the actual knowledge we have about the problem addressed.
Line 221: The correct conjugation is "have associated".
References:
Consider reviewing the references’ style formatting in order to grant uniformity (e.g. year of publication always in bold).
Consider updating your bibliography by adding at least a couple of reference published in 2018, which enrich the discussion. A possible paper to consider for quality evaluation of water/system in critical setting could be the following:
Troiano G., Messina G., Zanieri E., Li Donni V., Nante N., Magistri L., Pulci MB. Niccolini F.
Qualita’ microbiologica delle acque per emodialisi: quali i fattori di rischio?
[Microbiological quality of hemodialysis water: what are the risk factors?]
G Ital Nefrol., 2018, 5, 35:144-150
Consider, for the introduction, deleting reference 2 because reference 1 is enough.
Consider citing just a couple of references in place of 8-14.
Consider, for the introduction, deleting references 22 to 26 since reference 1 is enough.
Consider citing just a couple of references in place of 27-32.
Author Response

(The authors gave the same response as above.)

Round 2
Reviewer 1 Report
Please find comments attached.

Author Response
Dear Reviewer,
Please find in the attached file the second revision of the "ijerph-446008" manuscript; the Authors have made the changes in accordance with the referees' suggestions and answered to every question moved. The authors hope that in the current version the manuscript is clearer and more suitable for publication in the International Journal of Environmental Research and Public Health.

Reviewer 3 Report
Strengths
There are significant improvements overall in this version as compared with the previous.
English is fine, just a few flaws should be fixed.
Weaknesses
Introduction:
Line 44: Consider using the plural noun "spas".
Line 47: The correct conjugation, with "cases" as subject, is "haven't been published". Consider also adding "yet".
Materials and Methods:
Hospital structure:
Line 77: Consider using "we went into".
Line 85: Consider using "above ground", if you are not including the underground floors.
Otherwise, consider using "in total".
Line 86: Consider using the plural "Departments", and also using the more common term “clinics” in lieu of “ambulatories”.
Sampling:
Please consider describing more thoroughly the actual water collection procedures, describing in particular the temperature measuring operations and the precautions you took to avoid sample contaminations, or include a reference if you followed an already standardized procedure.
Isolation and serological identification of Legionella spp.:
Line 130: A comma is not supposed to be between “co.” and “Ltd.”.
Results:
Line 141: Be sure to delete the typo.
Line 160-161: With this wording, the reader understands that 72% of the total water samples were positive to L. pneumophila serotypes 2-14.
Consider writing the sentence like this: "Regarding positive water samples, the most of them were contaminated by L. pneumophila serotypes 2-14 (72%) whereas L. pneumophila serotype 1 accounted for the 18%.".
Lines 167-168: Be sure to delete the typo.
Discussion:
Line 190: Be sure to put the actual text in the correct position and to delete the typo.
Line 193: be sure to put the text in the correct position.
Author Response

(The authors gave the same response as above.)
